# An a Priori Approach to Small Airway Dysfunction in Pediatric Asthmatics

**DOI:** 10.3390/children9101454

**Published:** 2022-09-23

**Authors:** Russell Hopp, Junghyae Lee, Heather Bohan

**Affiliations:** 1Department of Pediatrics, University of Nebraska Medical Center, 42nd and Emile St., Omaha, NE 68198, USA; 2Children’s Hospital and Medical Center, 8200 Dodge St., Omaha, NE 68114, USA; 3Department of Biostatistics, College of Public Health, University of Nebraska Medical Center, Omaha, NE 68198, USA

**Keywords:** asthma, children, adolescents, small airway dysfunction

## Abstract

Small airway dysfunction remains a stepchild in the pediatric asthma care pathway. In brief, elements of the pulmonary function test (PFT) concerning smaller airway data remain less utilized. To further the value of the standard PFT we underwent a prospective Proof of Concept (POC) project, utilizing the outpatient performance of PFT tests in children 6–18 years during a 15-month period. The goal of the study was to determine if a priori the PFT represented a small airway disease pattern or not. Only the pulmonary function was used to make that distinction. Children 6–18 years with asthma who completed a PFT had their PFT as being characterized with or without a small airway dysfunction (SAD) designation, coded in the electronic medical record as an a priori decision using the code J98.4 (other disorders of lung) as a marker for electronic medical records retrieval. Subsequently, the results were analyzed between a group of 136 children designated (a priori) as having no small airway dysfunction in comparison to 91 children a priori designated as having small airway dysfunction. The a priori designation groups were post hoc compared for large and smaller airway function differences. Both large and smaller airway dysfunction were highly significantly different between the 2 groups, based solely on the initial division of the total group based on the decision the PFT represented a small airway pattern. We concluded the baseline pulmonary function test used in the evaluation of pediatric asthma has readily identifiable information regarding the presence of small airway dysfunction, and we characterized what was unique on the PFT based on that SAD classification

## 1. Introduction

A critical region of the lung, the small airways of pediatric asthmatics, can be a particularly difficult area to investigate and target with therapy. A recent review by Hopp et al. titled “ **Small airways disease in pediatric asthma: the who, what, when, where, why, and how to remediate. A review and commentary”** provides a wide-ranging discussion of the problem [1]. The review covers the time course of the recognition of the problem, a comparison of diagnostic possibilities and a discussion of potential therapies [1].

Recognized in the 1970′s as a “silent zone” of the lung, minimal attention is paid to any straightforward methodology of investigating its contribution to pediatric asthma [1,2,3,4]. Standard pulmonary function, however, provides a wealth of small airway data, which was outlined in reviews by Hopp et al. [1,5].

A 2014 review of techniques for assessing small airways dysfunction (less than 2 mm) reviewed information of spirometry, plethysmography, impulse oscillometry, single breath nitrogen washout, multiple breath nitrogen washout, helium and sulphur hexafluride washout, exhaled nitric oxide, high resolution computerized tomography (CT), hyperpolarized helium, magnetic resonance imaging (MRI), and nuclear medicine techniques of two-dimensional gamma scintigraphy, single photon emission computed tomography, and positron emission tomography [4]. It is clear from the above list that only spirometry is readily available, with exhaled nitric oxide a distant second, especially in children.

A consortium study of adult asthmatics published in 2019 revealed that the single measure of Forced Vital Capacity (FVC) improvement after albuterol, followed by the forced mid-expiratory flow (FEF_25-75_%) were both correlated to small airway dysfunction as defined by high-res CT [6]. Lung clearance was not nearly as sensitive [6]. 

A recent study in asthmatics found older, more obese, Type 2 inflammation and smoking to be related to both functional exercise capacity and diminished small airway measures of impedance oscillometry, FEF_25-75,_ and a lung clearance index [7].

Another recent study in adult asthmatics showed monitoring FEF_25-75_ has important outcome priorities [8]. Of particular interest to our report, they designed a FEF_25-75_ of less than 65% of predicted as their standard for stating a subject had airway dysfunction [8].

Other recent studies of asthmatic children have also focused on SAD, using traditional and less readily available technology [9,10,11].

To further focus the use of FEF_25-75_ in pediatric asthma outpatient medicine we present the results of a prospective analysis of small airway dysfunction in children initiated in April 2020 and completed in June 2021. All pulmonary function tests in pediatric asthmatics seen by a single asthma specialist (RJH) on those dates were included, with an analysis of differences in large and smaller airway function using only standardized spirometry.

The goal of the study was to determine if the presence of small airway dysfunction (SAD) was a priori and could be assigned as present (or not) based on the data available on the standard pulmonary function test. Small airway dysfunction was assigned based on a priori basis, with all elements of the PFT considered **prior to the group comparison analysis**. The investigator providing the assignment of the J code for SAD (RJH) had recently published an extensive review of small airway disease in children [1]. Bayesian probability is presumed to have properly sub-divided the total group into small and non-small airway dysfunction but the hypothesis of proper designation was tested with the post hoc statistical analysis.

## 2. Methods

### 2.1. Study Population

Asthmatic subjects 6–18 years were included in the analysis if seen between 1 April 2020 and 30 June 2021. The study was performed during the early and mid-COVID timeframe. All children seen by the investigator had to go to a single center, and testing was done by a limited cadre of respiratory therapists. Although impulse oscillometry was available, the institution’s COVID restrictions did not permit utilization during the study period. The subjects were all seen by one specialist. The PFT was restricted to a 30-minute room access between patients, and no additional testing was allowed (impulse oscillometry). The study was arbitrarily ended when COVID restrictions at the single center ended, and additional off-site testing sites were used by the investigator.

The visits included asthma follow-up or new asthma visits. The primary purpose of the small airway dysfunction assignment was for purposes of establishing the density of disease in the population of children seen in a referral center for asthma and was initiated as part of a proof of concept (POC) project following an extensive review of the topic [1]. The principal investigator is part of a Division of Pulmonary and Allergy and in an academic setting. The enrolled subjects underwent standard spirometry, with or without a post-albuterol assessment. The purpose of a post-albuterol test was for the identification of reversibility, assessment of therapy, or assisting with step-down or step-up decisions. When a post-albuterol was not done, it was generally due to stability of known asthma, without reason for medication adjustment. In large part, this was a real-world setting, but with a specific approach to SAD identification as a POC. All the children with a satisfactory PFT during the study period were retrospectively included in the analysis, as long as the visit was performed within the study dates allowed by the IRB. There was no power analysis performed as it was a retrospective analysis of a concept arising from the likelihood the investigator could identify SAD only from a PFT analysis.

We choose the top two measures found in the Postma and Qin protocols to be available and clinically relevant, FVC and FEF_25-75_ [6,8]. The FEF_25-75_ served as the principal surrogate in our study. However, other measures were included in the analysis. (Table 1).

The Institutional Review Board of the University of Nebraska Medical Center approved this study in 2021 (IRB#:0653-21-EX UNMC IRB) as a retrospective chart review study. Subjects were not consented at the inclusion of the J 98.4 code as the designation was done at the time of the visit as part of the clinical record, and the decision to do the retrospective analysis was decided post-visit and after the IRB had approved the chart review. The data analysis had names, and medical record numbers removed.

### 2.2. Pulmonary Function

Pulmonary function testing was completed for the pediatric population. The Vyaire Sentry Suite program was used for spirometry measurements, using nose clips and a mircroguard filter. Patient statistics were obtained including age, height, race, sex, and weight. The normative data used for this population was Global Lung Function Initiative (GLI) [12]. Baseline spirometry and post-bronchodilator spirometry was performed, if ordered. A short-acting beta 2 adrenergic agonist, Albuterol 4 puffs with spacing device, was used to obtain bronchodilator reversibility. Spirometry data was selected according to ATS criteria, however if the patient did not reach 6 s of exhalation, the results were deemed acceptable if a plateau of at least 1 s was reached by determination of time and patient effort/ability, and the test was considered acceptable in the view of the age of the subject. The printed report data included FVC, (Forced Expiratory flow in one second) FEV_1_, FEF_25-75_%, FEV_1_/FVC ratio, and Z-scores, and flow volume loop images.

### 2.3. Small Airway Assignment and Coding

The visit for PFT that occurred in the calendar year April 2020–June 2021 included an a priori decision for adding J code 98.4 for that visit. The principal investigator determined, based solely of PFT data if SAD was present. The J code was used as an electronic record marker (not used for any other purpose) for the a priori decision that SAD was present. The PFT was done and included for analysis on a day of asthma stability, and the data was assigned to this analysis for that day *only*, regardless of previous tests. Previous test(s), if done were not reviewed for trend or to influence of the code used on a specific day for the test (98.4-other disorders of the lung). Data on each subject was tabulated using the electronic medical record program EPIC. A paper copy of the complete PFT was available for downloading. Data were extracted using search criteria of asthma, ages 6–18, asthma, and date of visit April 2020–2021. The J code 98.4 had been added as present or absent for the assigned visit. Only 1 visit was allowed for inclusion, starting 1 April 2020.

In large part this was a real-world study, adding the unique concept of small airway dysfunction assignment based on a PFT over-view. The assignment was then verified using a statistical comparison of the SAD and non-SAD J 98.4 groups. The a priori positioning as SAD present or absent was the dependent variable and was clinically assigned prior to the analysis, and the data extracted from the PFT served as the independent variables that could be statistically tied to the pre-determined J 98.4 code (used to mark the a priori presence or absence of suspected SAD coded in the electronic medical record).

### 2.4. J 98.4 Code

The J code 98.4 was added to the International Classification of Diseases (ICD), Clinical Modification coding system in 2016, and was updated to the 2022 ICD-10 classification system. It is a code for “other disorders of the lung, or lung disease not otherwise specified”. It was used solely as a marker for the decision, based on the PFT, that small airway disease was apparent. It was a marker in the electronic medical record, along with the asthma code. It was used for retrieving those children with asthma and pre-determined SAD, along with those with asthma and no pre-determined SAD.

## 3. Statistical Analysis

Statistical analyses were performed using the SAS 9.4 software package [13]. First, all continuous primary indicators were tested normal distribution using Shapiro–Wilk, Kolmogorov–Smirnov, Cramer–von Mises, and Anderson–Darling tests [14]. We also checked descriptive figures including normal distribution table and QQ plots. A non-parametric statistical test was carried out using the Mann–Whitney U Test (Wilcoxon signed-rank test) to determine whether there is a difference between the levels of age, FEF_25-75_ percent predicted, Z-score FEF_25-75_ and the difference between the baseline FEV_1_–FEF_25-75_% predicted, ratio of baseline FEV_1_/FVC among these two groups [14]. These indicators were not normally distributed and are given as the median, minimum and maximum values. A parametric statistical test was carried out using the independent t-tests to determine a statistically significant difference between the means of FEV_1_/FVC ratio, FEV_1_ Z SCORE and FVC of these two groups, which were given as mean ± standard deviation. A FEF_25-75_ below the z score of −1.645 was compared between the two groups using a Fischer’s exact test [14] to negate the use of the FEF_25-75_ (at least in adults with COPD [15]). Comparison of the FVC between the two groups was done using a chi-square analysis [14].

If a post-bronchodilator study was done during the clinical decision pathway, the results were compared between the response in the non-SAD and SAD groups using a Chi-square analysis [14].

## 4. Results

For the 16 months of the POC project 227 children were included in the analysis if they performed a satisfactory PFT as part of their asthma visit in the clinic of the author (RJH). A post hoc statistical analysis was made of the a priori small airway dysfunction code (J 98.4) done on the day of the visit.

One-hundred thirty-six children were assigned with a priori determined as not having small airway dysfunction, and 91 children were assigned to the small airway dysfunction code. Table 1 presents the results of age, gender, baseline % predicted FVC, baseline FEV_1_ percent predicted, FEF_25-75_ percent predicted, z score for the FEF_25-75_ and the numerical difference between the FEV_1_ and the FEF_25-75_ percent predicted. A gender difference was seen, as many of the a priori SAD males were a segment of the authors long-standing clinic practice. As this was a real-world study, albuterol was used *if* a new patient or for medication step-up or step-down planning.

As seen in Table 1, the a priori decision, using J code 98.4 strongly supported both a small airway and a large airway difference. FEV_1_ precent predicted and the ratio of FEV1/FVC revealed large airway differences, while FEF_25-75_ and z score for FEF_25-75_ revealed smaller airway differences, both of a very significant nature. The unique use of product of the % predicted of FEV_1_ (-) FEF_25-75_ was to determine a proportional large airway and small airway numerical value, which was strongly statistically different between the two groups. The % predicted baseline FVC was not different between the groups, showing sustained volume, but obstructed lung function in the J98.4 group.

Using a cut-off of ≤65% for FEF_25-75_, none of the non-SAD (n = 136) had a value below the cut-off while 37 of the 91 SAD group did. The difference was significant at <0.00001. The mean for the SAD group for FEF_25-75_ was 66.9%.

For the possibility that a low FVC might negate the use of the FEF_25-75_ [15], the number of children below the −1.645 z score was determined. One child in the non-SAD group had an FVC z score of less than −1.645 while 3 of 91 children in the SAD group had a z score < −1.645, and the difference was not significant (*p* > 0.05). However, removing those 4 subjects in a sub-group analysis did not change the significance of the data analysis.

Finally, a limited number of subjects also had a post-albuterol test, 41 (28%) of the non-SAD group and 38 (42%) of the SAD group. As this was a real-world study, albuterol was potentially used *if* it was a new patient or for potential medication step-up or step-down planning. For the post bronchodilator study, pre-post differences in FEV_1_ (≥9%), FEF_25-75_ (≥35%) and FVC (≥10%) were compared. The non-SAD vs. SAD group had no difference in FVC improvement, but a highly significant difference for improved FEV_1_ (*p* < 0.00001) and improved FEF_25-75_ (*p* < 0.00001) in the SAD group.

## 5. Discussion

The presence of small airway dysfunction is a current topic of investigation in the adult population and has been recently published in a predominately adult population using functional characteristics of physical activity and symptom control [6,7,8]. Studies of this type can serve as a blueprint for prospective pediatric asthma investigation and potential earlier intervention, and are under-represented in the pediatric literature

A group of world-wide collaborators are actively small airway dysfunction (SAD) in adult asthma [6]. A study is listed at ClinicalTrials.gov and sponsored by Chiesi Farmaceutici S.p.A., with a title of “ **AssessmenT of smalL Airways involvemeNT In aSthma (ATLANTIS),** is a multinational, multicenter, non-pharmacological intervention, cross-sectional and longitudinal study” [6]. Their study goal is the discovery of a clinically relevant, easy to use methodology of determining SAD in adult asthmatics [6]. Although much debate surrounds the assignment of SAD, most centers have ready availability of PFT testing, but less so other measures. The study of Postma supported FEF_25-75_ as a reasonable surrogate for a CT of the chest [6]. Their model was used for this protocol, only using readily available results (PFT) during COVID.

In this project, however, we first attempted to determine the burden of SAD in a pediatric asthmatic population in tertiary care setting population. The secondary goal was to statistically prove our a priori assignment of SAD and non-SAD children was correct. For the fifteen-month period, all the children and adolescents in our academic clinical practice were seen at one speciality hospital, due to COVID limitations. For this time-period impulse oscillometry was not available, nor body-box plethysmography. Both might be valuable in SAD measurements, but as mentioned FEF_25-75_ and FVC are reasonable surrogates [6,8]. The sole physician assessing the patients, and presented in this study, used standardized pulmonary function results, obtained by hospital-based respiratory therapists. All subjects included were seen for routine asthma visits, or new asthma visits. The diagnosis of asthma had been previously determined (or was that day) and the known asthmatic subjects were on standard asthma therapy as defined by the NHLBI guidelines from 2007 or placed on that day based on severity [16]. In the case of known asthma, pulmonary function tests utilizing albuterol were only done when medication step-up or down was being considered. New asthmatics were enrolled when the PFT pre-post albuterol demonstrated >a 12% change in FEV_1_ along with the clinical decision the patient had asthma. Asthma medications were maintained or modified or started for each subject but are not delineated for this report. If nitric oxide was done, it was used as an independent variable to this report and was not included in the decision-making for the J 98.4 coding.

Once the children were assigned to SAD, the code J 98.4 was added as a separate asthma code in the electronic medical record. In part, the code was added for ease of analysis, but as seen in Table 1 the assignment demonstrated very significant implications. Tests of total lung capacity and large airway function, FVC, FEV_1_ and FEV1/FVC and z scores were analyzed.

The publication of Potsma et al. had suggested the FVC had a high predictability for a SAD phenotype (using chest CT as the gold standard) [6]. Another study has shown a change in FVC > 10% had high clinical relevance as a marker of SAD [17]. In this report there was not a significant difference in FVC between the two groups (Table 1). Only 4 children in the entire study had an FVC z score less than −1.645. Among the subjects undergoing post-albuterol, only four subjects in each group improved their FVC > 10%. These data strongly represent preservation of FVC even in the face of marked obstruction in large (FEV_1_) and smaller airways (FEF_25-75_) in the small airway disease group.

In the large airway tests, FEV_1_ and FEV_1_/FVC ratios were significantly different between the groups. This significant difference shows both a large and smaller airway obstruction component for the SAD group. There was a significant improvement in FEV_1_ for twenty five of the 38 post-albuterol tests in the SAD group, and only eight of the 41 no-SAD group (*p* < 0.00001). The FEV_1_/FVC ratio difference reflected the concomitant large airway obstruction in the SAD group (*p* < 0.00001).

The FEF_25-75_ was the value with the most clinical weight in distinguishing the two groups. The statistical analysis showed a highly significant difference (*p* = 4.5 × 10^−35^). Even without a post-albuterol test, the medication the subject was on at the time of the test was not sufficient, or in the case of a new patient indicated a higher step-care was needed. Equal to FEV_1_ improvement was the improvement in FEF_25-75_ in the thirty-eight who performed an albuterol challenge (25 improved) SAD group of >35% (*p* < 0.00001). Only 8 of 41 non-SAD had improvement in FEF_25-75_, but as a group they started at 108 ± 15%). Figure 1 shows the box plot of the FEF_25-75_. The boxplot suggests a FEF_25-75_ percent predicted less than 80% will be a good cut-off for strongly suggesting SAD.

The unique analysis of the product of the % predicted of FEV_1_-FEF_25-75_ was also strongly significant (Pulmonary Function Discrepancy). To our knowledge, this determination has not been used previously used, but provides a simple number to benchmark. A boxplot of this value (Figure 2) shows marked separation between the groups. A difference of 20 or more showed a large separation (at the 75 percentile) of the FEV_1_% predicted (-) FEF_25-75_% predicted. A larger prospective study would need to be done to further assess its applicability.

Small airway measures have had a less obvious position in pediatric asthma management [1]. This is not as true in adult asthma, as recent efforts are underway to determine characteristics of small airway disease in adults [6,7,8] and matched to their physiological findings [8]. A recent report of mismatch of FEF_25-75_ and FEV_1_ in adult asthma as a measure of airway dysfunction has been published, which used FEF_25-75_ as the marker of SAD [8]. In that study, the FEF_25-75_ was better than the FEV_1_ in predicting airway hyperresponsiveness and severe asthma. They used a FEF_25-75_ < 65% as marking small airway dysfunction. In our study, Thirty-nine of the 91 SAD children had a FEF_25-75_ less than 65%, while 74% had a FEF_25-75_ below 80%. So, in fact, the airways beyond the 7–8th generation (>2 mm) had less than an 80% capacity in the majority of the SAD group, with a mean FEF_25-75_ for the SAD group of 66.9%.

Our report mirrors this data [6,7,8], but in a pediatric population. In addition, a very recent study suggests impulse oscillometry (IOS) can be of further value in assessing large airway obstruction in adults [18]. Their protocol only investigated FEV_1_ and FVC baseline along with IOS testing. In our study, large airway measures plus more distal measures were compared. IOS was not included here, however, due to COVID restrictions during the study period.

SAD has been recently discussed in the pediatric asthma arena in two recent reviews [1,5]. Using information from the standard pulmonary function test we were able to show highly significant differences between the groups for smaller airway dynamics. FEF_25-75_ and the z score for FEF_25-75_ had markedly statistical differences (Table 1) and had minimal overlap between the two groups. The unique continuous variable of the predicted FEV_1_(-) FEF_25-75_ was also significantly different with minimal over-lap between the 2 groups. It is possible this number bridges the boundary of a strictly large vs. small airway caliber dysfunction. It has not had previous usage and needs perspective analysis.

There are limitations to this study. The asthmatics were all of one investigator, and of a more significant nature as they were seen at a tertiary care hospital for children. The parents had to be able to arrive with their children/adolescents during the COVID-time period. The children self-selected, based on an in-person appointment, or post-telehealth visit for their PFT. In large part, it was a real-world protocol. Finally, there is no gold standard for absolutely establishing small airway disease, except for a CT, which was beyond the scope of this report. A recent adult-based protocol provided a model for this analysis [6]. Other limitations include non-use of IOS or body-box plethysmography as supportive tests of SAD, but these tests require separate visits and were not allowed during the COVID period. Studies using IOS are useful in selected populations, but possibly less so when the base PFT answers most questions [18].

The strengths include a single investigator providing the a priori classification, and consistent well-trained respiratory technicians doing the PFT studies in a hospital-based laboratory. The resultant data mirrors the evolving literature on SAD [6,7,8,18,19]. This report provides a one-of-a-kind, real-world experience of determining the presence of smaller airway dysfunction using standard pulmonary function test analysis. IOS testing could add *further* value [18], but its use in our protocol was not included. In addition, suspecting SAD based on a pediatric PFT could allow for CT-scan or IOS support for SAD in a subsequent study, as is being done in young adults in an American-Lung Association-NHLBI sponsored prospective multi-centered study in young adults [20], and the international study of adults [6]. It may also serve as an indication for earlier step-up therapy, such as biologics.

## 6. Conclusions

In conclusion, the experience of this a propria determination of SAD emphasizes the ability to clinically analyze PFT data and determine SAD likelihood. Being able to do so obviates more complex testing and always for quicker disease categorization and medication management decision, as present published guidelines add no additional step-care assignment based on PFT smaller or IOS-detected small airway obstruction [21]. The concepts discussed here provide an approach to detecting early SAD that might be amendable to more aggressive therapy, but currently are under-emphasized in current guideline-based care [21].

## Figures and Tables

**Figure 1 children-09-01454-f001:**
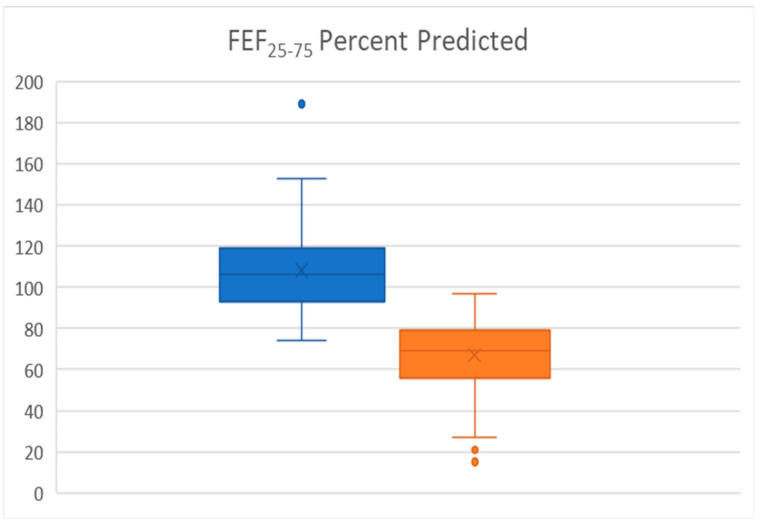
Box plot and 95% confidence limits for Non-SAD (blue) and SAD (orange) FEF_25-75_% predicted.

**Figure 2 children-09-01454-f002:**
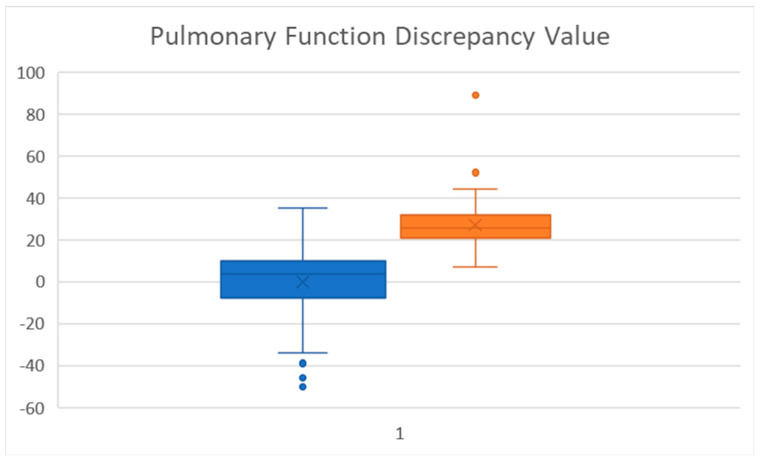
Difference between % predicted FEV_1_% Predicted FEF_25-75_ (Blue, No SAD, Orange, SAD).

**Table 1 children-09-01454-t001:** Comparison of a priori selected No SAD and SAD groups.

	No SAD	Mean ± SD	Median	SAD	Mean ± SD	Median	*p* Value
N	144			92			
Gender	76 M /68 F			61 M /31 F			0.004 *
Age		11.3 ± 3.4	11.0		11.1 ± 3.3	12.0	*p* > 0.05 ***
FVC % predicted		107 ± 12.6	106		105 ± 13.5	105.5	*p* > 0.05 ***
FEV_1_ % predicted		107.5 ± 11.7	106		93.9 ± 15.4	95.5	2.3 × 10^−13^ **
Z score FEV_1_		0.66 ±.97	0.59		−0.52 ± 1.06	−0.37	<0.00001 ***
Ratio FEV_1_/FVC		88 ± 5.2	88.5		77.2 ± 7.06	78.0	<0.00001 ***
% FEF_25-75_ baseline percent predicted		108 ± 19.6	106		66.9 ± 15	69	4.5 × 10^−35^ **
Z score FEF_25-75_		0.32 ± 0.81	0.22		−1.52 ± 8	−1.31	1.71 × 10^−25^ **
FEV_1_ percent predicted–FEF_25-75_ percent predicted		−0.05 ± 16	3.5		27 ± 10.9	25.5	1.63 × 10^−31^ **

* Chi square, ** Independent Wilcoxson test, *** t-test.

## Data Availability

Not applicable.

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
