# Peer review of "An a Priori Approach to Small Airway Dysfunction in Pediatric Asthmatics"

_children, 2022, doi:10.3390/children9101454_

Round 1

Reviewer 1 Report

The article is very interesting and addresses an important health problem in the child population.

I have some suggestions and technical notes that will improve your manuscrypt.

Explain more clearly the survey methodology.

Line 5-8 no address of the University, country

Line 24: no conclusion from the study performed in the abstract

Line 25: No keywords

Line 73: Please underline what is the aim of the study

The Methodology doesn’t contain information on the consent of the bioethical commission to conduct the study and the number with the date of issuing such an opinion

There is no information as to whether the parents or legal guardians have given their consent to participate in the study of children

No information is available on how the children were selected for the study? How the test sample was calculated?

Full experimental details must be provided so that the results can be reproduced.

Discussion: Please underline why this article is publishable, which is innovative.

Future research directions may also be mentioned. 

Introduction and discussion: no citations of bibliographic items visible in the text in accordance with IJERPH guidelines, e.g. [1]

No Conclusion separation. Please separate in the text Conclusion (line 317)

References should be written in accordance with IJERPH guidelines. The number of bibliographic items requires supplementing in the introduction and discussion part, only 17 bibliographic items. It is worth referring to the reviews on ,,Small-airway dysfunction in pediatric asthma” that have been published in recent years.

It is worth adding Abbreviations at the end

line 330: Institutional Review Board Statement: no data 

line 331: Informed Consent Statement: no data

line 332: Data Availability Statement: Acknowledgments: no data

Author Response

Thank you for your suggestions:  we responded below in bold.

I have some suggestions and technical notes that will improve your manuscrypt.

Explain more clearly the survey methodology.

We have added key elements to improve understanding

Line 5-8 no address of the University, country

Added

Line 24: no conclusion from the study performed in the abstract

Added

Line 25: No keywords

Added

Line 73: Please underline what is the aim of the study

Underlined in the Revision

The Methodology doesn’t contain information on the consent of the bioethical commission to conduct the study and the number with the date of issuing such an opinion

Added

There is no information as to whether the parents or legal guardians have given their consent to participate in the study of children

Added

No information is available on how the children were selected for the study?

Added and hopefully clarified

How the test sample was calculated?

No power analysis done, but based on the p value of differences, an adequate sample was obtained.  This was a post-hoc analysis of a data point isolated at each visit (J 98.4)

Full experimental details must be provided so that the results can be reproduced.

This is a totally different approach to a manuscript writing.  Some journals offer version of this, but this is the not approach we will take

Discussion: Please underline why this article is publishable, which is innovative.

We have added information on this in the revision, and underlined several sentences.

Future research directions may also be mentioned. 

We added several concepts

Introduction and discussion: no citations of bibliographic items visible in the text in accordance with IJERPH guidelines, e.g. [1]

Not certain the meaning of this.  Each section is labeled.

No Conclusion separation. Please separate in the text Conclusion (line 317)

We separated and added to the conclusion

References should be written in accordance with IJERPH guidelines.

It appears we have honored the style for Children:

MDPI and ACS Style

Vinante, E.; Colombo, E.; Paparella, G.; Martinuzzi, M.; Martinuzzi, A. Respiratory Function in Friedreich’s Ataxia. Children 2022, 9, 1319. https://doi.org/10.3390/children9091319

The number of bibliographic items requires supplementing in the introduction and discussion part, only 17 bibliographic items. It is worth referring to the reviews on ,,Small-airway dysfunction in pediatric asthma” that have been published in recent years.

We added several references of recent works

It is worth adding Abbreviations at the end

Not a standard way ?

line 330: Institutional Review Board Statement: no data 

added in body

line 331: Informed Consent Statement: no data

added in body

line 332: Data Availability Statement: Acknowledgments: no data

???

Reviewer 2 Report

1. Suggestion In the "No Sad" and "Sad" columns in Table 1, the "Age" and the following rows should give the "min-max" data.

2. P values for the comparison of age and FVC% prediction should be given.

3. The independent Wilcoxson test is more suitable for the non-parametric test of two independent samples of graded data, and also for the parametric test of two independent samples of non-normal distribution. If it is used to test the parameters of two independent samples with normal distribution, it will reduce the efficiency. From the comparison of Mean and Median, the AGE/FVC% predict/FEV1% Predict / % FEF 25-75 baseline data appear to be normally distributed measurements, but the Independent Wilcoxson test was used.

4. T-test is applicable to the measured data with normal distribution or normal distribution after treatment. From the comparison of means and medians, z-score FEV 1 data do not appear to be normally distributed, but a t-test was used.

5. Fischer's exact test is only used to test count data rates, and the description in lines 159-161 is clearly problematic.

Author Response

My co-author is a statistician and responded to the Reviewer's comments below.  We both stand on our analysis as described in the manuscript. 

Thank you for the comments. Before deciding parametric or non-parametric analytical model, the empirical distribution of the data was tested with descriptive and analytic approaches. First, the authors have checked histograms, normal probability plots, and quantile-quantile plots to see the data distributed normally or not.  Second, we also examined several normality tests, which are Kolmogorov-Smirnov test and Lilliefors test that show the estimated mean and variance from the data. Based on these statistical tests, analytical model for each indicator was built for either parametric or non-parametric methods. That is, if hypothesis of normality was rejected at the significant level of 0.05, Wilcoxon Signed Rank Test was applied (non-parametric approach), otherwise Independent Sample T-test was used (parametric approach).

His other comments:

1. Suggestion In the "No Sad" and "Sad" columns in Table 1, the "Age" and the following rows should give the "min-max" data.

The Columns for the SAD and NON-SAD were bolded.  This seems a style of Table suggestion, and I left it the same.  The mean and SD are listed which can be used to calculate a minimum and maximum.

2. P values for the comparison of age and FVC% prediction should be given.

> 0.05 and that was added

3. The independent Wilcoxson test is more suitable for the non-parametric test of two independent samples of graded data, and also for the parametric test of two independent samples of non-normal distribution. If it is used to test the parameters of two independent samples with normal distribution, it will reduce the efficiency. From the comparison of Mean and Median, the AGE/FVC% predict/FEV1% Predict / % FEF 25-75 baseline data appear to be normally distributed measurements, but the Independent Wilcoxson test was used.

(see above comment from our statistician)

4. T-test is applicable to the measured data with normal distribution or normal distribution after treatment. From the comparison of means and medians, z-score FEV 1 data do not appear to be normally distributed, but a t-test was used.

(See above comment form the statistician)

5. Fischer's exact test is only used to test count data rates

A very nice discussion of structural zeros in contingency tables is provided by West, L. and Hankin, R. (2008), “Exact Tests for Two-Way Contingency Tables with Structural Zeros,” Journal of Statistical Software, 28(11), 1–19. URL http://www.jstatsoft.org/v28/i11

As the title implies, they implement Fisher’s exact test for two-way contingency tables in the case where some of the table entries are constrained to be zero.

6.  and the description in lines 159-161 is clearly problematic.

Round 2

Reviewer 2 Report

Table 1 needs to be amended. Most of the data in the table are measurement data rather than count data, and each variable comes from the same sample, so there is no need to repeat the same sample size in a separate column. 144 or 92 are not measurements of the variables in each row, and the list method in the original text is not appropriate. You can refer to the following format modification:

Although the above table is from a published article, there are still some flaws. The next one is better. Maybe the “Range” and “Median” column are not obligatory, but they can make the content of the article more informative and add luster to the article.

Author Response

The Table was modified (RED) and the N=144 and N= 92 were removed in the columns.  The range was not included in the Table, as it overlaps with mean and standard deviation.  

"Even for distributions that are not normally distributed the bulk of the data will be within the plus or minus two standard deviations, so it should provide a rough estimate of most any distribution"

The Tile was changed for better clarity
